# A Systematic Review of Research on Non-Maternal Caregivers’ Feeding of Children 0–3 Years

**DOI:** 10.3390/ijerph192114463

**Published:** 2022-11-04

**Authors:** Tanya Nieri, Arianna Zimmer, Jai Mica Vaca, Alison Tovar, Ann Cheney

**Affiliations:** 1Department of Sociology, University of California Riverside, Riverside, CA 92521, USA; 2Center for Health Disparities Research, School of Medicine, University of California Riverside, Riverside, CA 92521, USA; 3Department of Justice Studies, San Jose State University, San Jose, CA 95192, USA; 4Department of Behavioral and Social Sciences, Brown University, Providence, RI 02912, USA; 5Department of Social Medicine, Population and Public Health, School of Medicine, University of California Riverside, Riverside, CA 92521, USA

**Keywords:** child feeding, non-maternal, caregivers, early childhood

## Abstract

Although people other than mothers participate in feeding, few interventions include non-maternal caregivers, especially those promoting healthy development among children aged 0–3 years. Understanding the role and influence of non-maternal caregivers is essential for the development and effectiveness of early childhood feeding interventions; yet, no reviews have examined non-maternal caregivers of children aged 0–3 years. This study assessed what is known about non-maternal caregivers’ feeding of children aged 0–3. We systematically reviewed 38 empirical quantitative, qualitative, and mixed methods studies, cataloged in PubMed and Web of Science and published between 1/2000–6/2021. The studies showed that non-maternal caregivers engage in child feeding and their attitudes and behaviors affect child outcomes. Like mothers, non-maternal caregivers vary in the extent to which their knowledge and attitudes support recommended feeding practices and the extent to which they exhibit responsive feeding styles and practices. Children of broad ages were included in the studies; future research should include infant/toddler-only samples to allow for better assessment of age-specific feeding constructs. The studies also revealed issues specific to non-maternal caregivers that are unlikely to be addressed in interventions developed for mothers. Thus, the review highlighted features of non-maternal caregiving of children 0–3 years that could be addressed to support feeding and child outcomes.

## 1. Introduction

While what infants and toddlers eat is related to their development and health [1], how infants and toddlers are fed also influences their outcomes [2]. Feeding styles (e.g., restrictive versus indulgent) and practices (e.g., force feeding, propping a bottle) relate to growth and obesity risk [3,4,5,6,7,8]. In particular, responsive feeding is recommended for anyone who feeds an infant or toddler because it is associated with healthy growth and lower obesity risk [9]. However, most research on feeding young children has focused on mothers [10]. Emerging evidence shows the importance of the role that non-maternal caregivers (NMCs) play in feeding [11,12]. NMCs are a broad category including diverse caregivers, such as fathers, grandparents, babysitters, and daycare workers. They may be primary or secondary caregivers. Because there is no consensus definition of NMCs, this review takes a broad view to understand what and who NMCs are and how they relate to the care of young children. 

While some research explores the role of NMCs among school-aged children [13,14,15,16], less research examines the role of NMCs among very young children (ages 0–3). It is important to understand the influence that NMCs may have on feeding in this group of children in order to develop appropriate and effective interventions that can promote healthy eating behaviors and prevent chronic disease. While some research examines how NMCs indirectly affect children by influencing mothers’ feeding [17,18,19], especially breastfeeding [11,20,21], there is a particular need to understand how NMCs directly affect children through their own feeding knowledge, attitudes, and practices. The present study aims to answer these questions: What is known about NMCs’ feeding of children aged 0–3? To what extent is NMCs’ feeding similar to or different from mothers’ feeding?

## 2. Materials and Methods

This systematic review follows the Preferred Reporting Items for Systematic Reviews and Meta-Analyses (PRISMA) guidelines [22] (see Appendix A). The first three authors identified which publications were eligible for review and what data should be extracted from the publications. All authors provided input on the results and discussion.

### 2.1. Inclusion Criteria and Search Strategy

We included published original empirical articles. The articles had to address care given to children aged 0–3 years by NMCs. We focused on this age group given the limited research in this area and the developmental stages it captured: infancy and toddlerhood. We included articles that provided information on non-maternal caregiving or information on the effect of NMCs on feeding-related child outcomes (e.g., growth). We included studies with a variety of study designs (qualitative or quantitative, cross-sectional or longitudinal, etiological or intervention). Thus, we included articles if they provided information on the topic even if the main research questions were not about NMCs’ feeding–for example, we included a study whose main research question was about the effectiveness of a cash transfer program because it provided information on feeding knowledge and practices [23]. We included articles with older children if they also included children 0–3. We included articles that provided information on NMCs whether the data came from the caregivers themselves or another source, such as mothers. We included articles regardless of the country in which the research was conducted. The articles had to be published between January 2000 and June 2021, be in either English or Spanish (since we are a bilingual team), and be referenced in the databases PubMed and Web of Science. We excluded literature reviews, study protocols, and non-empirical articles. We also excluded articles whose topic did not address feeding-related caregiving or NMCs and articles in which the study sample did not include children in the target age range of 0–3 years. 

For our PubMed search, we employed both Medical Subject Heading (MeSH) terms and free keywords to identify articles (see Appendix A). This search returned 722 abstracts. For Web of Science, we employed a similar search with Boolean operators. This search returned 366 abstracts. We screened the articles to remove duplicates within each database and across databases and remove articles that did not meet the eligibility criteria. This process left us with 38 articles for inclusion in the review. Figure 1 shows the flow of articles into the review.

### 2.2. Analysis

The analysis first involved a summary of each study’s key characteristics. Table 1 contains each study’s citation and a description of the design and methods, the sample, and the feeding constructs assessed. Second, the analysis of study findings aimed to identify the types of NMCs, NMCs’ feeding knowledge, attitudes, behaviors, and styles, factors related to NMC feeding, differences between mothers and NMCs, and the relation of NMCs to children’s outcomes. Third, we assessed each study’s design and methods for the risk of bias. Consistent with recommendations by Johnson and Hennessey [24], we used existing evaluative frameworks, described in Petticrew and Roberts [25]: one for the quantitative research (evaluated in Appendix A) and one for the qualitative research (evaluated in Appendix A). Both sets of criteria were applied to mixed methods studies. To determine whether the studies met the appraisal criteria, we employed the following rating system: Good, Fair, and Poor. All studies earned a Good or Fair rating and were, thus, retained in the review.

## 3. Results

### 3.1. What Is a Non-Maternal Caregiver?

Because the studies varied in their definition of an NMC, the types of caregivers included in the studies varied greatly. Ten studies provided information on how “caregiver” was defined [33,34,35,36,41,53,57,62,63,64]. Six studies defined caregivers broadly, to include people involved in feeding and other care [33,35,36,41,57,63]. Four studies defined caregivers narrowly, to include people directly involved in feeding children [28,34,53,62]. A majority of the studies defined the mother as the primary caregiver, and anyone else as a secondary caregiver. The exceptions to this rule were seven studies, all carried out in Asia, that examined NMCs who were primary caregivers [23,27,41,42,53,57,63]. 

### 3.2. Who Are Non-Maternal Caregivers?

Wasser et al. [62] and Chung et al. [33] provided comprehensive information on who provides care for the child; all other studies did not. The common approach was to focus on a single category of caregivers. Eight studies focused exclusively on fathers [26,37,40,45,48,50,54]. Five studies focused exclusively on grandparents [33,38,42,44,57]. Seven studies focused exclusively on formally employed child care workers [30,34,36,46,56,58,61]. One study focused on students in training to become child care workers [49]. One study included multiple NMC types: fathers, grandmothers, and child care workers [62]. 

The remaining 16 studies included NMCs and mothers. Nine examined the two types of caregivers for similarities and differences [28,29,31,35,43,47,52,59,60,63]. The remaining seven studies examined the two types of caregivers as one group [23,27,32,39,41,53,55]. 

Of the 15 studies including grandparents, whether alone or with other caregivers, seven included grandfathers [23,28,38,42,44,47,57]; the rest included only grandmothers [32,33,35,43,52,53,62,63]. 

Two studies enumerated the people providing care to a child. Wasser et al. [62] examined whether any NMCs were used and if so, into what category of NMC the caregivers fell. They found that more than half of the households used an NMC, the most frequent categories of whom were fathers, grandmothers, and child care workers. Chung et al. [33] asked mothers to enumerate the people who were involved in care, but only among people living in the home. It then focused on details of the care provided by grandmothers.

### 3.3. How Do NMCs Perceive Their Role and/or Responsibility in Feeding?

Seven studies provided information on NMCs’ perceived role and/or responsibility in feeding. Love, Walsh, and Campbell [49] found that child care worker trainees described feeding as an important part of their role as a child care worker and expressed a desire to be a positive role model for children. Blissett et al. [31] found that mothers reported greater feeding responsibility than fathers, with 62% of fathers reporting being seldom responsible for feeding and 3.2% of fathers reporting never being responsible for feeding. 

Five of the seven studies focused exclusively on fathers. Guerrero et al. [37] found that about 40% of fathers reported having a great deal of influence on their preschool child’s nutrition and about 50% reported daily involvement in feeding. Khandpur et al. [45] found that 62% of fathers reported sharing feeding responsibilities with the child’s (2–10 years) mother, and among the remaining fathers, about half reported being solely responsible for feeding and the other half reported being not at all responsible for feeding. Similarly, Mallan, Nothard et al. [50] found that many fathers perceived that they were responsible at least half of the time for feeding their child in terms of organizing meals (42%), amount of food offered (50%), and deciding if their child eats the “right kind of foods” (60%). Mallan, Daniels et al. [51] found that fathers who were more concerned about their child becoming overweight reported higher perceived responsibility for child feeding. Fathers with higher work hours, lower socioeconomic status, and younger children perceived less responsibility for feeding. Fathers’ body mass index and education level and child gender were not associated with perceived responsibility. Lindsay et al. [48] found that Brazilian immigrant fathers perceived a greater role in feeding, relative to fathers in Brazil, since their immigrant status meant they had fewer people to help with child care in the U.S. They also found that fathers who had experienced food insecurity in the past perceived a greater role in feeding. 

### 3.4. To What Extent Are Non-Maternal Caregivers Involved in Feeding?

The studies provided sparse details on the extent of feeding involvement, such as time commitment, making it difficult to distinguish across studies between NMCs playing a major role and NMCs playing a minor role in caregiving (e.g., occasional caregiving), especially in the case of secondary caregivers. Three studies provided information on what predicted the extent and nature of involvement. Mallan et al. [50] found that fathers were more likely to be involved in feeding if they spent less time in paid employment, held higher perceived responsibility, and believed fathers should be involved with their children. Roshita et al. [55] found that NMCs’ specific feeding responsibilities depended on whether the NMC coresided with the mother and child. Coresidence meant that the mother chose and bought the food and the extended family member who was the NMC prepared it, whereas in the absence of coresidence the NMC was likely to also choose and purchase the food. This study also found that the feeding role of domestic paid caregivers was broader if the relation with the family was longer standing. Finally, the study found that when mothers lacked feeding-related self-confidence, specifically lack of confidence that they could cook, they relied more heavily on NMCs for feeding. Chung et al.’s [33] study, using a global measure of caregiving that included feeding and other tasks, found that mothers reported that paternal grandmothers, compared to maternal grandmothers, did more caregiving and that grandmother involvement did not differ by child sex. 

One study predicted use/non-use of an NMC by mothers. Wasser et al. [62] found that employed mothers were more likely to use an NMC. Married mothers were more likely to involve fathers as caregivers and less likely to involve grandmothers as caregivers. Younger mothers were more likely to involve grandmothers. Mothers with older infants (12- and 18-month-olds) were more likely than mothers of younger infants (3-month-olds) to use child care workers. 

### 3.5. How Do Non-Maternal Caregivers Relate to Mothers?

Six studies provided information on the relations between mothers and NMCs, points of disagreement between caregivers, and the way decision making is handled when multiple caregivers are involved. Lidgate et al. [47] found that parents perceived there to be cross-generation conflict in feeding practices. Parents described a battle between current feeding recommendations and the previous experiences and opinions of older NMCs in the family. Although parents looked to their own parents for feeding advice and support, such as related to breastfeeding and the introduction of solid foods, they perceived their own parents to have outdated opinions and to pressure them to act on those opinions.

Jiang et al.’s [42] study of grandparents as primary caregivers found that grandparents who had experienced poverty experienced limited decision-making power in a three-generation home due to their reliance on the parents of the child for accommodations. Thus, in cases where their feeding preferences differed from those of the parents, they were less likely to act on their preferences so as to avoid conflict. However, the researchers also found that the grandparents experienced their caregiving to be a heavy responsibility, one they did not want to do poorly. This sense of responsibility was related to overfeeding, as the grandparents perceived a child’s accumulated body fat as a sign of successful feeding and care. 

Eli et al.’s [35] found that when there was disagreement about feeding among mothers and maternal grandmothers, most grandmothers deferred to the mothers’ decisions. However, some grandmothers claimed a “grandparent’s prerogative” to indulge grandchildren when they spent time with them. Some mothers ignored the grandmother’s indulgence if the child spent only a small amount of time with the grandparent.

The remaining three of the six studies focused on fathers. Anderson et al. [26] found that fathers and male partners of mothers perceived themselves as an assistant to the mother who was the primary caregiver and that while they may discuss feeding with the mother, they showed deference to her. Horodynski and Arndt [40] found that some fathers, who were actively involved in and committed to feeding, felt that their ways of feeding their child were not respected by their partners and they wanted their child to learn the father’s ways. These African American fathers described the existence of a cultural perception that African American fathers should not be actively involved in or committed to child care, including feeding. Khandpur et al. [45] found cooperative feeding practices–that is, practices that fathers reported to be concordant between mothers and fathers–in about half of the cases in their sample and conflicting feeding practices in 40% of the cases. Mothers and fathers tended to agree on having food rules, providing a non-distracting eating environment, and monitoring the child’s food intake. Conflicting practices related to access to unhealthy snacks and dietary variety and were driven by differences in parental eating habits, feeding philosophies, and concern for child health. About equal numbers of married and divorced/separated fathers reported conflicting practices. Feeding disagreements were resolved typically through discussion and negotiation of a compromise and, less commonly, through consultation with health care providers. 

### 3.6. What Do Non-Maternal Caregivers Know about Feeding?

Nine studies provided information on NMCs’ knowledge about feeding [23,26,36,39,41,43,46,53,57]. The studies varied widely in the types of knowledge about which they asked, making summary and comparison across studies difficult. However, as research on mothers [65,66] has shown, these studies indicate that NMCs may have knowledge that is inconsistent with recommended feeding practices. No clear pattern emerged across the studies about which specific knowledge NMCs are likely to have consistent with feeding recommendations. Only Karmacharya et al.’s [43] study directly compared the knowledge of NMCs to that of mothers. It found that mothers’ feeding knowledge was greater than grandmothers’ knowledge. 

Two studies provided information about NMCs’ sources of information about feeding. Anderson et al. [26] found that fathers received feeding information from health care practitioners, their prior experience with children, printed materials, and other people who have children. Yue et al. [63] found that grandmothers were less likely than mothers to refer to official sources for feeding information.

Two studies addressed the relation of feeding knowledge to behavior. Freedman and Alvarez [67] found that knowledge was not always congruent with feeding behavior. Lanigan [46] found that improvements in child care workers’ knowledge was significantly correlated with improvements in feeding practices.

### 3.7. What Are Non-Maternal Caregivers’ Feeding Attitudes?

Six studies provided information on NMCs’ feeding attitudes [39,40,42,46,48,53]. The studies varied widely in the attitudes about which they asked, making summary and comparison across studies difficult. However, as found in research on mothers [17,68], these studies showed that feeding attitudes were associated with feeding practices [46] and that NMCs may have problematic feeding attitudes, such as the belief that heavy children are healthy children [42,53]. 

### 3.8. What Are Non-Maternal Caregivers’ Feeding Practices?

Twenty-seven studies provided information on NMCs’ feeding practices (see Table 2). The specific behaviors assessed varied widely, making comparison across studies difficult. Some studies employed existing standards for feeding practices, such as those of the Institute of Medicine [30], UNICEF, US-AID, the World Health Organization [63]. Other studies involved an inductive approach in which the investigators described and summarized the practices observed in their sample. As with the findings on NMC knowledge and attitudes, the research in the review shows that NMCs’ feeding practices include those that are not recommended. This finding is consistent with research showing that mothers employ practices that are not recommended [4,5].

One study identified funding and training as facilitating greater utilization of supportive feeding practices by child care workers [56]. Specifically, child care workers in centers that participated in the Child and Adult Care Food Program reported practices more consistent with a supportive feeding environment relative to child care workers in centers that did not participate. This difference was attributed to the training that the child care workers in funded centers received. 

Five studies empirically compared NMCs’ practices to mothers’ practices. Blissett et al. [31] found that mothers reported higher monitoring of the child’s food intake than fathers, regardless of the child’s gender, but mothers and fathers had similar reports of a tendency to restrict foods or pressure their children to eat. They also found mothers’ and fathers’ behavior was correlated in terms of the use of restriction and pressuring, but not in terms of the use of monitoring. Eli et al. [35], focusing on practices associated with beverage consumption, found that relative to mothers, grandmothers expressed less concern about setting rules for feeding but tended to follow their daughters’ lead in practice. Metbulut et al. [52] found that grandmothers, relative to mothers, were more likely to feed to soothe a child but were less likely to encourage food balance and variety, monitor child food intake, restrict food for health and weight reasons, and teach about nutrition. They found no difference in practices between maternal grandmothers and paternal grandmothers. 

Vandeweghe et al. [59,60] found that relative to parents and non-parental family caregivers, child care workers were less likely to offer children rewards for eating. They also found that while all three groups acknowledged that supportive feeding was possible, the groups differed in terms of the perceived feasibility of some feeding practices. For example, some non-parental family caregivers and most child care workers described having little control over the visual presentation and sensory sensations of food. Non-parental family caregivers indicated that their choice of what to buy and cook may be limited by the budget set by the parents. Child care workers reported that the kitchen staff, not them, prepare the food and determine its presentation. Child care workers also acknowledged the importance of offering food variety but reported that they do not typically decide the menu; rather it is determined by a designated nutritionist who does not participate in the children’s meal time. Child care workers and non-parental family caregivers described other limitations on their ability to enact ideal practices. Institutional restrictions and limited staffing capacity, for example, limit child care workers’ ability to engage children in food preparation and serving. Non-parental family caregivers, because they typically operate alone and without the assistance of other people, reportedly find modeling by eating alongside the children to be challenging because they are attending to multiple children and tasks at once. These results are consistent with findings by Love et al. [49], who did not formally compare NMCs to mothers but found that parents’ expectations for feeding in a child care setting may be inconsistent with child care workers’ training and the setting’s policy.

Yue et al. [63] found no differences between grandmothers and mothers in terms of the timely introduction of complementary feeding and the feeding of staple food. However, they found that mothers were more likely than grandmothers to feed vegetables, fruits, and meat and attend to vitamin intake. They explained these differences in terms of caregiver education level and source of feeding information; relative to grandmothers, mothers had more education and were more likely to get their information from official sources. 

One study empirically compared feeding practices among NMCs. Tan et al. [57] found that relative to grandparents who were primary caregivers, grandparents who were secondary caregivers were much less likely to set a maximum limit on the amount of unhealthy food eaten and to offer a wide variety of food. 

### 3.9. What Are Non-Maternal Caregivers’ Feeding Styles?

Feeding styles refer to sets of attitudes (e.g., crying interpreted as meaning an infant needs to feed) and behaviors (e.g., encouraging an infant to finish a bottle) that characterize caregivers’ approaches to maintaining or modifying children’s eating behaviors. According to recent feeding guidelines for infants and toddlers, a responsive feeding style, as opposed to other styles, is recommended [9]. Responsive feeding involves properly interpreting and responding to the child’s hunger and satiety signals. 

Two studies provided information on feeding styles, measured quantitatively. In their study of mothers and fathers of infants in the U.S., Benjamin-Neelon and Neelon [29] used the Infant Feeding Style Questionnaire (IFSQ) [69] which measures five styles, one of which is the responsive style. They found that fathers scored higher than mothers on the pressuring and laissez-faire styles and maternal and paternal feeding styles were correlated. Mothers and fathers did not differ on the responsive feeding style.

Barrett et al. [28] also used the IFSQ and found that mothers’ and NMCs’ feeding styles differed. Compared with mothers, grandparents scored lower on the laissez faire and indulgent (permissive) styles; fathers scored higher on the pressure (soothe) and indulgent styles; and child care workers scored higher on the restriction style (diet quality) and responsive styles (satiety). Barrett et al. also identified caregiver and infant characteristics that were associated with feeding style. Being a male caregiver, being an obese caregiver, and residing in the child’s household were associated with feeding style. Infants’ weight-for-length z-scores, level of fussiness, and age older infants were associated with feeding style.

### 3.10. What Are the Effects of Non-Maternal Caregivers on Children?

Seven studies examined the relation of NMCs to children’s outcomes. Five of these studies showed that NMC characteristics relate to NMC feeding and children’s growth, although the mechanisms used to explain how this occurs varied widely. Tovar et al. [58] found that supportive feeding practices were associated with a higher quality diet among children. Guerrero et al. [37] found that when fathers reported eating out with their child a few times a week, compared to rarely or never eating out, children had over two times the odds of consuming fast food at least once a week. Fathers’ report of eating out with their children was also associated with children’s sweetened beverage intake.

In their longitudinal study Chung et al. [33]) found that grandmothers’ greater involvement in caregiving, measured at three months postpartum, was positively associated with the child’s 2-month weight z-scores but negatively associated with 24-month weight z-scores. Grandmothers’ greater involvement in caregiving, measured at 12 months postpartum, was associated with improved 12-month cognitive and fine motor skills and 24-month socioemotional development. No associations were found for length z-scores. This study measured caregiving to include feeding, but did not isolate the effect of feeding; therefore, the outcomes cannot definitively be attributed to the grandmothers’ feeding behavior.

Two studies examined the effect of grandparent co-residence on children. He, Li, and Wang [38] found a positive effect of grandparents’ co-residence with the child on child weight. The authors explained this effect as occurring through changes in the child’s fat intake (grandparents overfeeding the child) and physical activity (grandparents limiting the child’s activity due to risk aversion), although they did not specifically measure overfeeding or risk aversion. For that matter, they did not measure the grandparents’ caregiving behavior generally or feeding specifically. The effect on fat intake existed more in urban areas than in rural areas, and the effect on physical activity existed more in rural areas than in urban areas. There were no gender differences in these two outcomes for kids under 6 years old. Katzow et al. [44] found that persistent grandparent co-residence with the child from birth (versus none) was associated with higher child mean weight-for-age z-scores (WFAz) and higher odds of child overweight/obesity risk at 2 years and 3 years. It was also associated with feeding a bottle with cereal in it. Intermittent grandparent coresidence (versus none) was associated with higher odds of excessive juice intake, but persistent grandparent coresidence (versus none) was not associated with the odds of excessive juice intake. Cereal in the bottle and excessive juice consumption did not mediate the relation of grandparent co-residence to child WFAz. As with He et al. [38], this study did not measure NMCs’ feeding.

Two studies had results suggesting a relation between NMC involvement and child outcomes, but their designs did not allow for a definitive conclusion. Ansuya et al. [27] found that when caregivers gave prelacteal food or restricted foods, their children were more likely to be underweight. However, since this study did not separate mothers from NMCs and included so few NMCs (*n* = 11/570 primary caregivers), the relative impact of NMCs is unclear. Wasser et al. [62] found that utilization of an NMC for caregiving was longitudinally associated with an increased likelihood of infants and toddlers consuming juice or whole fruit and a decreased likelihood of breastfeeding. They found no association between NMC involvement in caregiving and early introduction of complementary foods or amount of food consumed by the child. This study did not capture who did the feeding; therefore, the outcomes cannot be directly connected to the NMCs’ feeding behavior. 

### 3.11. Assessment of Study Design

When interpreting the results of the studies in the review, it is important to consider their research designs and assess bias. Table 1 summarizes the design features of each study in the review. Of the 38 studies 23 were quantitative, 13 were qualitative, and two were mixed methods. Of the quantitative studies 10 were longitudinal. The cross-sectional nature of the remaining quantitative studies precludes claims of causal relationships between variables.

There is substantial diversity in the samples analyzed. Eighteen studies included U.S. samples; 20 fielded samples from other countries, whether developed or developing. Some studies involved rural samples, others involved urban samples, and some involved both or did not indicate. Similarly, some studies specifically focused on poor or low-income people or on low-resource geographic areas. The sample diversity complicates the ability to compare across studies, particularly to compare findings for specific measures of feeding constructs. For this reason, we limited our conclusions to the broad constructs of knowledge, attitudes, practices, and styles and found that, across the samples, there is room for intervention with NMCs to address their feeding. With regard to sample size, while the qualitative studies generally did not pursue generalizability, several quantitative studies employed small samples, leading the investigators to characterize their studies as exploratory and qualify their results as preliminary (see Appendix A). 

While our focus was on NMCs’ feeding of children aged 0–3 years, many of the studies included a broader range of ages. Only 13 of the studies focused exclusively on infants and/or toddlers. Some of the remaining studies included children as old as 12 years. The age of the children in a study likely influenced the investigators’ choice of feeding constructs and, in turn, shaped the results. For example, propping a bottle is a feeding practice for younger children whereas having family mealtimes is a feeding practice for older children. Future research should consider inclusion of only infants and toddlers in order to generate age-specific feeding results and ultimately, allow for a comparison across studies with similar samples. The specific target population, geographic context, and target age of children needs to be considered when developing interventions to ensure maximal responsiveness.

Regarding the source of data analyzed, nearly all of the studies collected data from NMCs. However, four studies collected data about NMCs from mothers [33,44,62] or the household [38]. Maternal reports may not accurately capture all of NMCs’ caregiving activities and may be vulnerable to social desirability bias and a desire to avoid conflict in interpersonal relationships [33]. Several studies collected data about children from mothers [52,62]. For example, Metbulut et al. [52] used mothers’ report of child feeding problems. NMCs’ evaluations of the children may differ from mothers’ evaluations. Thus, it would be ideal to capture data directly from NMCs to understand their role and influence. Two other ways that the samples are diverse is whether they included multiple types of NMCs and whether they included mothers as a comparison group.

With regard to measures of NMC feeding, some studies employed measures designed for mothers [28,29,34] or measures that were not validated because they were newly developed [39,57]. Dev et al. [34] adapted and validated for child care workers a feeding practices measure originally designed for parents. Barrett et al. [28] found some feeding style measures, designed for mothers, to have low reliability among grandparents and child care workers. The use of unvalidated measures may have influenced the study results.

Only Karmacharya et al. [43] reported a potential conflict of interest. The study was funded by an organization implementing an intervention, but the analysis was not of the intervention. Thus, we concluded that the study was not biased.

## 4. Discussion

This systematic review summarized and evaluated the extant literature on NMC feeding of children aged 0 to 3 years. The findings are summarized in Table 3.

### 4.1. Commonalities between Mothers and NMCs

Few of the studies compared mothers and NMCs. Those that did found some differences in feeding but provided no clear pattern of difference. Furthermore, many of the findings in the studies of NMCs were consistent with findings in research on mothers. Given our findings, there is not sufficient evidence to conclude that there are no differences between mothers and NMCs. However, the results suggest that mothers and NMCs draw on similar sets of strategies. They also support the broad conclusion that like mothers, NMCs vary in the extent to which their knowledge and attitudes support recommended feeding practices and the extent to which they exhibit responsive feeding styles and practices. Thus, intervention could be employed to engage and support NMCs, especially given the clear evidence from the review that NMCs participate in feeding. Toward this end, scholars and practitioners need to broaden their focus beyond mothers to examine the multiple caregivers bearing on child feeding and develop and implement interventions that address NMCs, whether separately or in tandem with mothers. Given the similarities with mothers, building on existing interventions for mothers and adapting them for NMCs could be an efficient strategy.

### 4.2. Unique Features of NMCs’ Feeding

Although more comparative research is needed, the extant literature did not reveal any particular NMC type to have especially problematic feeding relative to other types. That said, the review identified unique features of NMC feeding that should be considered when intervening with NMCs: decision-making power, cultural beliefs, and NMCs’ specific caregiving role. The review revealed how NMCs’ power relative to that of mothers or others shapes feeding in various ways. One aspect of power is resources. The amount of socioeconomic power an NMC has relative to mothers appears to influence feeding. An NMC’s reliance on mothers/parents for basic needs (e.g., housing, food) [42] can influence whether the NMC pursues their own feeding strategy or defers to the mother. Thus, even if an NMC may be familiar with a recommended strategy, they may not be able to implement it without jeopardizing their basic needs, if the mother prefers a different strategy. 

Related to socioeconomic status is child care resources. When a mother is heavily reliant on an NMC for child care, she may have (or feel she has) less power over feeding and thus, be likely to defer to the NMC on feeding matters, even if she disapproves of the NMC’s feeding strategies [47]. Alternatively, an NMC, recognizing that their care is a substitute for parental care, may engage in non-recommended feeding practices to compensate for the situation, aiming to show the child that they are loved [42]. Finally, the size of mothers/parents’ child care networks can shape NMCs’ perceptions of their role in feeding. Smaller networks and, in turn, fewer child care resources, such as in some immigrant communities [48], may translate to a perception of greater responsibility for feeding among NMCs than would be the case with a larger network with greater child care resources. Thus, the amount of child care resources can operate to constrain or facilitate whether and how NMCs engage in feeding. 

Aside from resources, power may exist in the form of decision-making authority. NMCs’ power to enact optimal feeding may be constrained by the conditions under which they perform their work–that is, the caregiving setting. This finding was salient in the case of child care workers. Vandeweghe et al. [59,60] and Wallace et al. [61] showed how NMCs cannot enact some supportive feeding practices because they do not control all of the decisions in and conditions of the caregiving setting. Furthermore, Love et al. [49] found that child care workers must consider parental wishes even if they are inconsistent with center policy.

In addition to power, this review revealed that cultural beliefs relate to NMCs’ feeding in various ways. In the case of grandparents as NMCs, a cultural belief in a grandparent prerogative–the notion that grandparents have the right to indulge their grandchildren–relates to grandparents’ feeding [35]. In terms of their roles vis a vie the grandchildren, some grandparents perceive a division of labor in which parents are in charge of nutrition and grandparents are in charge of entertainment. Thus, even when they are the designated caregiver of a child and responsible for feeding the child, they view themselves as exempt from feeding rules or guidelines and engage in indulgent feeding. A contributing factor is a belief in elder privilege in some countries, such that grandparents feel entitled to make decisions independent of the parents’ wishes and parents feel compelled to defer to grandparents out of respect or themselves view grandparent indulgence as appropriate [47]. The number of children in the family [42] and the NMC’s extent of involvement in caregiving [35,57] relate to the salience of this influence on feeding, highlighting how culture can operate differently in different caregiving contexts. 

Cultural beliefs also operate in the case of fathers. As exemplified in Horodynski and Arndt’s study [40] of African American fathers, cultural stereotypes rooted in structural racism, such as about the role of African American men and fathers, may constrain whether and how NMCs engage in feeding their children. This finding is consistent with research on American mothers documenting how American slavery influences contemporary breastfeeding practices through the cultural stereotype of the slave mammy [70]. 

Finally, a third unique feature of NMC feeding is the specific caregiver role. This review found that whether the NMC is a primary or secondary caregiver is relevant to feeding. While the results of this review do not provide great detail on the differences and similarities between primary and secondary NMCs, since so few studies captured information on the extent of NMCs’ caregiving involvement, there is enough information to suggest that attention should be paid to this issue. For example, this review showed that NMCs who are secondary caregivers may view themselves as helpers, and their desire to help mothers may result in feeding practices which may help the mother (e.g., bottle feeding to allow the mother to sleep through the night) or may help themselves as caregivers (introducing solid foods earlier than recommended so that all children in the family can be fed together) but are less ideal for the child [47]. It also showed that the relations between NMCs and mothers, such as whether there is concordance between them and how discordance is navigated, also shape feeding.

### 4.3. Implications for Research and Practice

Given the newness of the focus on NMCs, we employed highly inclusive criteria for the review. Yet, our search strategy and choice of databases may not have identified all of the existing research on the topic. A limitation of this review is the difficulty in comparing the studies due to the heterogeneity of the feeding constructs measured and the settings in which the studies were conducted (i.e., resource-rich and resource-poor settings). Another limitation is associated with the cross-cultural generalizability of the findings, given the diversity of samples and countries in the review studies. Cultural and socioeconomic differences in racial/ethnic groups within and across countries may differentially relate to NMC feeding patterns. 

The existing studies include broadly defined NMCs and narrowly defined NMCs. Future research needs to better and more consistently define what an NMC–more specifically, what an NMC involved in feeding–is. Furthermore, few studies included multiple types of caregivers. Comparisons between caregiver types could improve understanding of the multiple influences on a single child and understanding of similarities and differences between mothers and NMCs and between types of NMCs. To understand a single child’s outcomes, an ideal study design would include all the child’s caregivers. To inform interventions for specific target groups, an ideal design would include multiple caregivers to allow for group comparisons.

A related point is that research should define and examine primary and secondary NMCs. We expect that primary caregivers will have significant influence on children, but we need more studies to compare maternal primary caregivers to non-maternal primary caregivers. For example, this research could examine fathers as primary caregivers rather than as helpers for mothers. It could also examine non-maternal caregiving in same-sex parent families. The review showed that NMCs may serve as primary caregivers and in turn, potentially influence both the child and the child’s mother. Whether focusing on primary or secondary NMCs, studies need to separate in analysis mothers from NMCs, rather than lump them together. 

We also need more research on the nature and extent of secondary NMCs’ influence and how it may differ depending on the extent of involvement in feeding. For example, among secondary NMCs, an occasional NMC, such as a babysitter, may feed differently than a regular NMC. Because their caregiving is non-normative, an occasional NMC may be more likely to define the feeding occasion as a special one and in turn, deviate from recommended feeding guidelines, even in the absence of a special occasion (e.g., a birthday). We agree with Wasser et al. [62] that scholars should employ continuous rather than dichotomous measures of feeding involvement to better identify mechanisms of effect. Extent of involvement could be measured in terms of time and/or feeding tasks. Some NMCs are involved with food purchasing, preparation, and feeding while others are just involved with feeding.

Better definitions will facilitate research enumerating the types of NMCs caring for a single child and describing a child’s food ecology. They will also facilitate comparisons between caregivers broadly and NMCs specifically and between different food ecologies–that is, sets of caregivers. This research will further elucidate unique factors influencing NMC feeding and the conditions influencing their expression. Furthermore, a focus on food ecologies will allow for better understanding of the relations between caregivers, since their combined efforts produce child outcomes. Thus, rather than focusing on what any one NMC does, it is important to examine what an NMC thinks, believes, and does in the context of other caregivers. This research can then inform intervention tailoring for specific target groups.

Regarding design, longitudinal quantitative studies will help to illuminate causal relations and allow for assessment of how relations between non-maternal caregivers’ feeding and child outcomes change across stages of child development. Future quantitative research should validate feeding measures with large samples of NMCs, including multiple caregiver types [28]. Future research, whether qualitative or quantitative, should aim to include more narrow samples–that is, samples limited to children aged 0–3 years–to ensure that age-specific feeding is addressed. It could also explore samples of children with health or developmental problems or otherwise living in atypical situations. Other than the study of left-behind children in China [23], the studies in this review did not address such children. Future studies, especially quantitative ones, should employ larger samples to expand generalizability and/or allow for sub-group comparisons. Finally, they should gather data directly from NMCs rather than relying on proxy reports. 

Regarding implications for practice, intervention to promote healthy growth and prevent obesity in early childhood is sorely needed to address early childhood obesity, the prevalence of which has doubled in the United States (USA) from the 1970s to the 21st century [71,72]. Few existing interventions include NMCs [10], and those that do tend to apply lessons learned from mothers. Although more research on NMCs is needed (see next section), this review has highlighted not only the possibilities for intervention among NMCs but also NMC-specific issues that could be addressed in intervention research. 

The extant research on NMCs indicates a need to focus on children’s feeding ecology: moving beyond the mother-child dyad and attending to all the caregivers who are involved in feeding. An intervention does not necessarily have to include mothers and NMCs together or different types of NMCs together. However, it must address the issues that arise through the sharing of feeding responsibility for a given child and aim to cultivate a shared understanding of optimal feeding among caregivers. A focus on the food ecology through intervention also means accounting for culture’s influence on the roles each caregiver plays and the beliefs informing those roles. Intervention participants could reflect on how their role as a caregiver intersects with other social roles, such as being a father or a grandparent, and how it may shape their feeding. Lastly, it means accounting for power dynamics to understand whether and how the caregivers work together to feed the child. Interventions should foster communication between caregivers, teaching them to navigate discordance and cultivate concordance related to feeding. Towards these ends, intervention developers need to engage NMCs in intervention development to ensure that the approaches reflect their realities and are feasible for and amenable to them.

## 5. Conclusions

In conclusion, this review documented the important role of NMCs in feeding. It highlighted issues that NMCs share with mothers: deployment of a similar set of feeding strategies and variations in the extent to which knowledge, attitudes, practices, and styles conform to feeding recommendations. These issues can be addressed in interventions adapted for NMCs. The review also highlighted unique features of NMC feeding–issues of power, cultural beliefs, and the caregiving setting–which can be addressed in intervention but may require the development of new intervention content.

## Figures and Tables

**Figure 1 ijerph-19-14463-f001:**
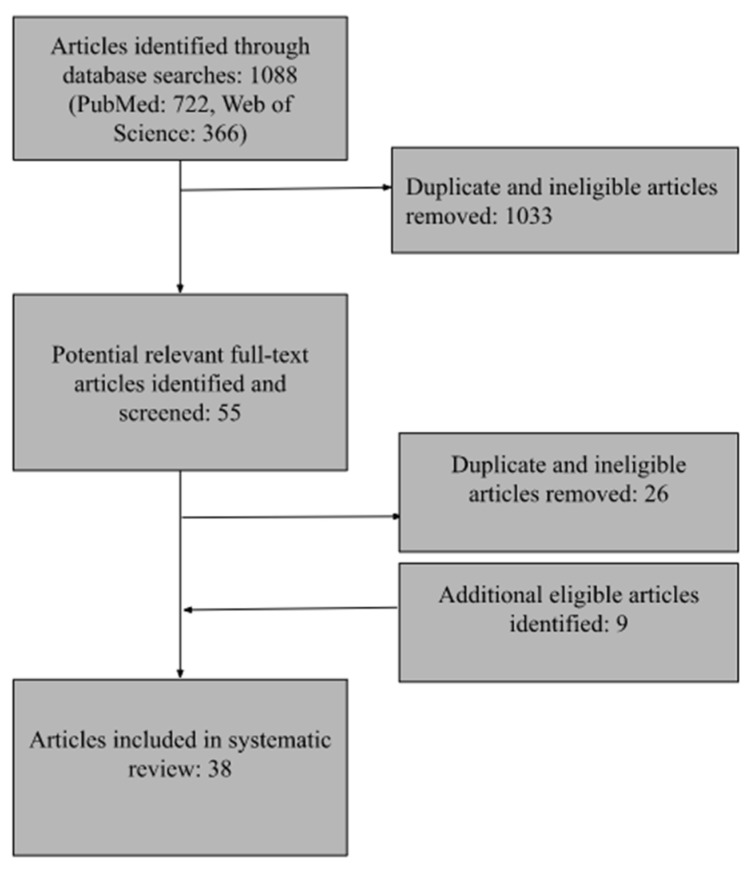
Flow of studies into review.

**Table 1 ijerph-19-14463-t001:** Summary of characteristics of studies in the review.

Study	Design and Method	Sample	Feeding Constructs Assessed
Anderson, Nicklas, Spence and Kavanagh [26]	Interview Qualitative	21 fathers or male partners of mothers of WIC-income-eligible infants 0–6 months old. Tennessee, U.S.	Feeding knowledge and practices Source of feeding knowledge Relations with mother
Ansuya et al. [27]	Cross sectional survey Quantitative	570 dyads: primary caregivers (mothers or NMCs) and their children 3–6 years old. Rural Karnataka, India.	Feeding practices Child outcomes of NMC involvement
Barrett, Wasser, Thompson and Bentley [28]	Longitudinal survey Quantitative	217 mother-infant dyads and 118 NMCs of the infants in the Women, Infants, and Children program. North Carolina, U.S.	Feeding styles
Benjamin-Neelon and Neelon [29]	Longitudinal survey Quantitative	202 families (mothers, fathers, and children 6–12 months old). Boston, MA, U.S.	Feeding styles
Blaine et al. [30]	Cross sectional survey Quantitative	166 Head Start child care workers (57 working with infants, 109 working with toddlers). Boston, MA, U.S.	Feeding practices
Blissett, Meyer and Haycraft [31]	Cross sectional survey Quantitative	188 cohabiting mothers and fathers of 94 children aged 12–62 months. Birmingham, Coventry, and Cambridge, United Kingdom.	Perceived role/responsibility in feeding Feeding practices
Chakona [32]	Cross sectional survey and focus group Mixed methods	Survey: 84 dyads: mothers or NMCs and their children 0–24 months old. Focus group: 94 mothers and grandmothers. Rural South Africa.	Feeding practices
Chung et al. [33]	Longitudinal survey Quantitative	1154 mothers of 3-month-old infants. Rural Pakistan.	Who are NMCs Predictors of extent and nature of feeding Child outcomes of NMC involvement
Dev, McBride, Speirs, Donovan and Cho [34]	Cross sectional survey Quantitative	118 Head Start child care workers serving children 2–4 years old. Urban Midwestern U.S.	Feeding practices
Eli, Hörnell, Etminan Malek and Nowicka [35]	Interview Qualitative	11 dyads: mothers and grandmothers of children 3–5 years old. Eugene, OR, U.S.	Feeding practices Relations with mother
Freedman and Alvarez [36]	Cross sectional survey Quantitative	72 child care workers serving children 6 months–5 years old. Eugene, OR, U.S.	Feeding knowledge and practices Relation of knowledge to practices
Guerrero, Chu, Franke and Kuo [37]	Longitudinal survey and interview Quantitative	2441 families: mothers, fathers, and their children 24 months old who lived with the biological mother. Nationally representative U.S. sample.	Perceived role/responsibility in feeding Child outcomes of NMC involvement
He, Li and Wang [38]	Cross sectional survey Quantitative	15,054 households with children ages 2–13 years. China.	Child outcomes of NMC involvement
Horodynski, Hoerr and Coleman [39]	Longitudinal survey Quantitative	38 low-income dyads: caregivers (mother, father, or grandmother) and their child 12–36 months old. Rural midwestern U.S.	Feeding knowledge, attitudes, and practices
Horodynski and Arndt [40]	Focus group Qualitative	6 African American fathers of children 1–2 years old, enrolled in Early Head Start. Jackson, MI, U.S.	Feeding knowledge and practices Relations with mother
Hossain et al. [41]	Focus group Qualitative	81 mothers or NMCs (fathers and paternal grandmothers) caring for children 6–59 months. Mirpur (urban) and Mirzapur (rural), Bangladesh.	Feeding practices
Jiang et al. [42]	Interview Qualitative	12 parents and 11 grandparents (4 grandfathers, 7 grandmothers) caring for children 3–6 years. Urban Beijing, China.	Feeding attitudes and practices Relations with mother
Karmacharya, Cunningham, Choufani and Kadiyala [43]	Cross sectional survey Quantitative	Mothers, grandmothers, and financial heads of household (typically fathers) from 4080 households with children 6–24 months. Rural Nepal.	Feeding knowledge
Katzow, Messito, Mendelsohn, Scott and Gross [44]	Longitudinal survey Quantitative	267 low-income, Hispanic mother-infant pairs. New York, NY, U.S.	Child outcomes of NMC involvement
Khandpur, Charles and Davison [45]	Interview Qualitative	7 fathers of children 2–10 years. United States.	Perceived role/responsibility in feeding Relations with mother
Lanigan [46]	Longitudinal survey and observation Quantitative	72 Head Start child care providers serving children 3–5 years. United States.	Feeding knowledge, attitudes, and practices Relation of knowledge to practices
Lidgate, Li and Lindenmeyer [47]	Focus group Qualitative	7 parents, 7 NMCs (non-parental family caregivers or informal child care workers) of children between ages 0–5. Edinburgh and Birmingham, United Kingdom.	Feeding practices Relations with mother
Lindsay et al. [48]	Interview Qualitative	21 Brazilian-immigrant fathers of children 2–5 years. Massachusetts, U.S.	Perceived role/responsibility in feeding Feeding attitudes and practices
Love, Walsh and Campbell [49]	Focus group Qualitative	19 students training to become child care workers serving children 2–5 years. Australia.	Perceived role/responsibility in feeding Feeding practices
Mallan et al. [50]	Cross sectional survey Quantitative	436 fathers of 2–5 year olds. Australia.	Feeding practices Perceived role/responsibility in feeding Predictors of extent and nature of feeding
Mallan et al. [51]	Cross sectional survey Quantitative	436 fathers of 2–5 year olds. Australia.	Perceived role/responsibility in feeding Feeding practices
Metbulut, Özmert, Teksam and Yurdakök [52]	Cross sectional survey Quantitative	200 children 2–5 years, 200 mothers, and 50 grandmothers. Turkey.	Feeding practices
Rachmi, Hunter, Li and Baur [53]	Focus group Qualitative	94 primary caregivers (mothers or grandmothers) of children 0–12 years. Greater Bandung Area, Indonesia.	Feeding knowledge, attitudes, and practices
Reisz et al. [54]	Longitudinal survey and observation Quantitative	118 first-time fathers and their 8-month old infants. Greater Austin, TX, U.S.	Feeding practices
Roshita, Schubert and Whittaker [55]	Interview Qualitative	26 mothers and 18 NMCs of children 1–3 years. Depok, Indonesia.	Feeding practices Predictors of extent and nature of feeding
Sigman-Grant et al. [56]	Cross sectional survey Quantitative	203 licensed child care center directors and 567 child care workers serving children 3–5 years old. California, Colorado, Idaho, and Nevada, U.S.	Feeding practices
Tan et al. [57]	Cross sectional survey and interview Mixed methods	Grandparents of children aged 12 years and below. Interview: 11. Survey: 396. Singapore.	Feeding knowledge and practices
Tovar et al. [58]	Observation Quantitative	Child care workers from 133 family child care homes serving children 0–5 years. Rhode Island and North Carolina, U.S.	Feeding practices
Vandeweghe et al. [59,60]	Focus group Qualitative	14 parents, 9 family child care providers, and 10 child care workers serving children < 6 years. Belgium.	Feeding practices
Wallace, Lombardi, De Backer, Costello and Devine [61]	Ethnography: internet forum and interview Qualitative	Interview: 42 child care workers serving children 0–5 years. Internet forum comments: 1179. Australia.	Feeding practices
Wasser et al. [62]	Longitudinal survey Quantitative	217 low-income, African American mother-infant dyads. United States.	Who are NMCs Predictors of extent of NMC use Child outcomes of NMC involvement
Yue et al. [63]	Cross sectional survey Quantitative	1383 infant caregivers (grandmothers and mothers) living in poor counties. Rural China.	Source of feeding knowledge Feeding practices
Zhang et al. [23]	Longitudinal survey and interview Quantitative	447 caregivers of left-behind children 3–5 years old. China.	Feeding knowledge and practices

**Table 2 ijerph-19-14463-t002:** Supportive and unsupportive feeding practices examined by studies in the review.

Studies Examining the Practice	Supportive Feeding Practice Examined
Anderson et al., (2010) [26]; Blaine et al., (2015) [30]; Freedman and Alvarez (2010) [36]; Lindsay et al., (2020) [48]; Love et al., (2020) [49]; Tovar et al., (2019) [58]	Attend to hunger satiety cues or allow children to leave food unfinished
Horodynski et al., (2004) [39]	Require child to be seated at mealtime
Blaine et al., (2015) [30]; Lanigan (2012) [46]; Lindsay et al., (2020) [48]; Mallan et al., (2014) [51]; Roshita et al., (2012) [55]; Sigman-Grant et al., (2003) [56]; Tovar et al., (2019) [58]; Wallace et al., (2020) [61]	Sit with children at meals or offer family-style meal
Blaine et al., (2015) [30]; Chakona (2020) [32]; Eli et al., (2017) [35]; Horodynski et al., (2004) [39]; Tovar et al., (2019) [58]; Yue et al., (2018) [63]	Offer fruits and vegetables
Blaine et al., (2015) [30]; Horodynski and Arndt (2005) [40]; Lindsay et al., (2020) [48]	Limit fast foods
Blaine et al., (2015) [30]; Eli et al., (2017) [35]; Lindsay et al., (2020) [48]; Metbulut et al., (2008) [52]; Tan et al., (2019) [57]	Limit sugary foods
Zhang et al., (2018) [23]; Chakona (2020) [32]; Eli et al., (2017) [35]; Horodynski et al., (2004) [39]; Yue et al., (2018) [63]	Serving a protein such as meat, eggs or milk
Freedman and Alvarez (2010) [36]; Lindsay et al. (2020) [48]; Mallan et al., (2013) [50]; Roshita et al., (2012) [55]; Vandeweghe et al., (2016) [59,60]	Employ mealtime routines (e.g., set meal time; rules such as no tv/electronics at table)
Horodynski et al., (2004) [39]; Lanigan (2012) [46]; Sigman-Grant et al., (2003) [56]; Tovar et al., (2019) [58]; Vandeweghe et al., (2016) [59,60]	Encourage tasting of foods
Horodynski and Arndt (2005) [40]; Lanigan (2012) [46]; Love et al., (2020) [49]; Tovar et al., (2019) [58]; Vandeweghe et al., (2016) [59,60]	Encourage or allow self-feeding
Sigman-Grant et al., (2003) [56]; Vandeweghe et al., (2016) [59,60]	Provide child-size tableware
Love et al., (2020) [49]; Metbulut et al., (2008) [52]; Sigman-Grant et al., (2003) [56]; Tovar et al., (2019) [58]	Discuss or teach about food/nutrition at mealtime
Lindsay et al., (2020) [48]; Love et al., (2020) [49]; Tovar et al., (2019) [58]; Vandeweghe et al., (2016) [59,60]	Model healthy eating
Vandeweghe et al., (2016) [59,60]	Attend to visual presentation and sensory characteristics of food
Lanigan (2012) [46]; Love et al., (2020) [49]; Vandeweghe et al., (2016) [59,60]	Engage child in food preparation or service
Horodynski et al., (2004) [39]	Avoid offering foods with choking risk
**Studies examining the practice**	**Unsupportive feeding practice examined**
Anderson et al., (2010) [26]; Metbulut et al., (2008) [52]	Feed to soothe
Anderson et al., (2010) [26]	Respond to food refusal by offering alternative foods
Anderson et al., (2010) [26]; Lidgate et al., (2018) [47]; Yue et al., (2018) [63]	Introduce solid foods early
Zhang et al., (2018) [23]; Freedman and Alvarez (2010) [36]; Horodynski et al., (2004) [39]; Jiang et al., (2007) [42]; Rachmi et al., (2017) [53]; Tovar et al., (2019) [58]; Vandeweghe et al., (2016) [59,60]	Tailor food offerings to child’s preferences, or not introduce new foods
Blissett et al., (2006) [31]; Love et al., (2020) [49]; Mallan et al., (2014) [51]; Metbulut et al., (2008) [52]; Reisz et al., (2019) [54]	Monitor child’s food intake
Blissett et al., (2006) [31]; Jiang et al., (2007) [42]; Lanigan (2012) [46]; Lindsay et al., (2020) [48]; Love et al., (2020) [49]; Sigman-Grant et al., (2003) [56]; Tovar et al., (2019) [58]	Pressure children to eat
Freedman and Alvarez (2010) [36]; Lanigan (2012) [46]; Tovar et al., (2019) [58]	Coach children to eat foods perceived as appropriate
Ansuya et al., (2018) [27]; Blissett et al., (2006) [31]; Dev et al., (2014) [34]; Eli et al., (2017) [35]; Love et al., (2020) [49]; Sigman-Grant et al., (2003) [56]	Restrict foods
Jiang et al. (2007) [42]	Serve large portion sizes
Eli et al., (2017) [35]; Hordynski and Arndt (2005) [40]; Jiang et al., (2007) [42]; Lanigan (2012) [46]; Lindsay et al., (2020) [48]; Love et al., (2020) [49]; Tovar et al., (2019) [58]; Vandeweghe et al., (2016) [59,60]	Offer rewards for eating or used food as a reward
Horodynski et al., (2004) [39]; Lidgate et al., (2018) [47]	Offer snacks
Eli et al., (2017) [35]; Lidgate et al., (2018) [47]	Indulge the child
Lidgate et al., (2018) [47]; Roshita et al., (2012) [55]	Employ or encourage bottle or formula feeding
Tovar et al., (2019) [58]	Eat sugary or salty food or beverage in front of child

**Table 3 ijerph-19-14463-t003:** Summary of findings.

Commonalities between Mothers and NMCs
They draw on similar sets of feeding strategies
They vary in the extent to which their knowledge and attitudes support recommended feeding practices
They vary in the extent to which they exhibit responsive feeding styles and practices
Intervention could engage and support NMCs
Maternal interventions could be adapted for NMCs
**Unique features of NMCs’ feeding**
*NMC decision-making power* vs. *that of parents or others:*
NMCs may not engage in recommended feeding if it jeopardizes their basic needs Mothers who heavily rely on NMCs may not challenge NMCs’ practices even if they are inconsistent with recommendations NMCs may engage in non-recommended feeding practices to compensate for parents’ inability to engage in feeding Families with smaller child care networks may rely more heavily on NMCs for caregiving Child care workers’ ability to comply with feeding guidelines may be constrained by other authorities, such as child care center food preparation logistics and parents’ expressed wishes for their children
*Cultural beliefs related to NMCs’ feeding:*
Belief in a grandparent prerogative is related to feeding indulgence Belief in elder privilege is related to deference to grandparents, even when feeding does not follow recommended guidelines Belief in fathers as not responsible for feeding constrains whether and how fathers engage in feeding
*Primary versus secondary caregiver roles:*
Secondary caregivers may prioritize helping over feeding guidelines
**Implications**
*For research:* Need a better and more consistent definition of NMC Need study samples with multiple types of NMCs to enable comparisons Examine role and impact of primary versus secondary caregivers Examine role and impact of a child’s set of caregivers (mothers and NMCs) Examine extent of NMC involvement, measured continuously not dichotomously Need longitudinal designs and larger samples Need samples focused on children 0–3 years to address age-specific feeding
*For practice:* Go beyond mother-child dyad to address child’s feeding ecology which includes NMCs Tailor interventions to address NMC-specific issues

## Data Availability

Not applicable.

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
