# Peer review of "A Systematic Review of Research on Non-Maternal Caregivers’ Feeding of Children 0–3 Years"

_ijerph, 2022, doi:10.3390/ijerph192114463_

Round 1

Reviewer 1 Report

In general, there is a need for some corrections and also a revision of the text because we identified English errors that impair the reading and the reader's understanding. The text needs to be clearer and more cohesive, there is little connection between ideas in the introduction and discussion.

Abstract
- It is important to mention the types of studies that were included in the review.

Methods
- No search strategy was identified for each database, that is, it was not verified how the Boolean descriptors were organized to perform the search.
- What were the criteria for including these databases? Why were searches in more usual databases such as Embase and Cochrane Library not performed?

- Although the inclusion criteria are clear in Table 1, the exclusion criteria are not well identified. What were they?

About the quality of the included studies: what was the checklist used

it would be interesting to work with the new update of PRISMA to build the flowchart of study selection 

Results: 

the way result section is written is more like a narrative review, i suggest the authors to tabulate the results, with the author, type of study, year, location, and outcomes

all the best. 

Author Response

We thank the reviewer for their comments. Our reply is in italics below. We have incorporated the feedback into the paper.

In general, there is a need for some corrections and also a revision of the text because we identified English errors that impair the reading and the reader's understanding. The text needs to be clearer and more cohesive, there is little connection between ideas in the introduction and discussion.

We reviewed the manuscript to correct any English errors. As we indicate below, this systematic review did not focus on a single outcome. Because the research topic (feeding by non-maternal caregivers of children 0-3 years old) is new, we addressed multiple questions: what types of NMCs exist, what are NMCs’ feeding knowledge, attitudes, behaviors, and styles, what factors relate to NMC feeding, what are the feeding-related differences between mothers and NMCs, and what is the relation of NMCs to children’s outcomes. We introduce these topics in the manuscript’s introduction and review them, with implications for future research and practice in the manuscript’s discussion.

Abstract
- It is important to mention the types of studies that were included in the review.

We edited the abstract to clarify that only empirical quantitative, quantitative, and mixed methods studies were included in the review.

Methods
- No search strategy was identified for each database, that is, it was not verified how the Boolean descriptors were organized to perform the search.

The search string is provided in Table S2: Search string for PubMed. On page 3/22 of the manuscript, we indicate that we used the same strategy for both databases.

- What were the criteria for including these databases? Why were searches in more usual databases such as Embase and Cochrane Library not performed?

We did not include the Cochrane Library because it focuses on intervention trials. Our topic was on factors relating to non-maternal caregivers’ infant feeding. We were not examining interventions to shape caregiver behaviors or infant feeding. We did not include Embase because it is highly similar to PubMed. We view our two databases, both of which have broad reach, to be sufficient. We modified the discussion section to acknowledge that our choice of databases shaped our results.

- Although the inclusion criteria are clear in Table 1, the exclusion criteria are not well identified. What were they?

We excluded literature reviews, study protocols, and non-empirical articles. We eliminated articles that did not meet the eligibility criteria (e.g., child's age was out of range, topic did not address feeding-related caregiving or non-maternal caregivers). We edited the manuscript to consolidate this information into one place on page 3/22. We did not include the details in Figure 1 to conserve space.

About the quality of the included studies: what was the checklist used

            The following supplemental tables provide info on the criteria for evaluating the included studies: Table S3: Evaluation of design and methods for quantitative studies in the review. Table S4: Evaluation of design and methods for qualitative studies in the review. These criteria came from Petticrew and Roberts which we cite on page 4/22. Here is the full citation:

Petticrew, M.; Roberts, H. Systematic reviews in the social sciences: a practical guide; Blackwell Pub.: Malden, MA; 2006.

it would be interesting to work with the new update of PRISMA to build the flowchart of study selection 

Our checklist is provided in Table S1: Index of information in the review meeting PRISMA guidelines. We agree that it would be interesting in future research to employ the new PRISMA. However, it should be noted that the new and the old versions are highly similar. We do not believe use of the new version would have significantly modified our results. We thank the reviewer for informing us of the update.

Results: 

the way result section is written is more like a narrative review, i suggest the authors to tabulate the results, with the author, type of study, year, location, and outcomes

We believe the format recommended by the reviewer would be ideal if we limited our review to quantitative studies where the predictors and “outcomes” are always clearly delineated. However, we include quantitative, qualitative, and mixed methods studies; thus, the variables vary greatly. Furthermore, as we explain in the methods section, we included studies if they provided empirical information on NMC feeding-related caregiving, even if the study’s main research questions were about something else. We did not want to confuse our outcomes of interest with the study’s outcomes. We made this choice to be as inclusive as possible, given that this area of research is new. The current Table 1 includes all the info requested by the reviewer, but the study findings. It is unclear how to add this info to the table given current space constraints. In the original submission we presented in a supplementary table (Table S5: Supportive and unsupportive feeding practices examined by studies in the review.) a summary of a key portion of the findings – the supportive and unsupportive feeding practices of non-maternal caregivers. Given the reviewer’s feedback, we moved this table into the main body of the manuscript. We also added Table 3 which accompanies the Discussion section.

Reviewer 2 Report

This paper talks about how non-maternal caregivers (NMCs) affect young children (age 0-3). Since NMCs’ feeding are related to children’s growth physically and mentally, it is very important to know how different factors are related. In this study, authors reviewed certain papers that focus on this specific topic and summarized results and ideas nicely and provided full-scale discussions. Overall, readers who have interests in this field or related fields will be benefited from reading this paper. However, I still have 3 major suggestions for this work:

1)    In your introduction, please provide readers a brief idea about NMCs by discussing more on this topic, such as, in urban/suburban area, NMCs are in demand to a certain percentage of young couples under certain ages. Something like that will explain the role of NMCs and emphasize the significance of the paper. Please also mention in the introduction that how NMCs are not interpreted clearly, or do not have a clear definition; maybe list some examples such as nursing workers, fathers, and grandparents, and mention that the influences from NMCs should be considered from different aspects, such as the jobs, cultural beliefs, and so on, which are related to what you are going to talk about next. 

2)    For the Result and Discussion sections, a lot of information are provided, and data are discussed in a very comprehensive way, which is good. However, it would be nice to have a table and briefly summarize all the information collected from different papers. Just like what you did for Discussion, split things in categories and then discuss. Having a clear table with information in order will help readers to follow the main ideas. 

3)    It is also nice to do a good summary about the paper in the Conclusion section. Expand this section, such as the paper highlighted what issues? What unique features? 

Author Response

We thank the reviewer for their comments. Our reply is in italics below. We have incorporated their feedback into the paper.

This paper talks about how non-maternal caregivers (NMCs) affect young children (age 0-3). Since NMCs’ feeding are related to children’s growth physically and mentally, it is very important to know how different factors are related. In this study, authors reviewed certain papers that focus on this specific topic and summarized results and ideas nicely and provided full-scale discussions. Overall, readers who have interests in this field or related fields will be benefited from reading this paper. However, I still have 3 major suggestions for this work:

  • In your introduction, please provide readers a brief idea about NMCs by discussing more on this topic, such as, in urban/suburban area, NMCs are in demand to a certain percentage of young couples under certain ages. Something like that will explain the role of NMCs and emphasize the significance of the paper.

We are unable to comply with this request as such universal information does not yet exist. If it existed, we would have included it in the review. We cannot say where NMCs are more likely to be located (e.g., urban or suburban areas), or that they are common among young couples. Our review did not reveal this information. The rationale for the review, which we state in the introduction, is that there is evidence that NMCs are involved in feeding. Thus, it is important to understand how they are involved in feeding and what impact this involvement has on children. Our review provides this understanding.

Please also mention in the introduction that how NMCs are not interpreted clearly, or do not have a clear definition; maybe list some examples such as nursing workers, fathers, and grandparents, and mention that the influences from NMCs should be considered from different aspects, such as the jobs, cultural beliefs, and so on, which are related to what you are going to talk about next.

We revised the introduction to incorporate the information recommended by the reviewer.

  • For the Result and Discussion sections, a lot of information are provided, and data are discussed in a very comprehensive way, which is good. However, it would be nice to have a table and briefly summarize all the information collected from different papers. Just like what you did for Discussion, split things in categories and then discuss. Having a clear table with information in order will help readers to follow the main ideas. 

We added a table (Table 3) summarizing the findings. 

  • It is also nice to do a good summary about the paper in the Conclusion section. Expand this section, such as the paper highlighted what issues? What unique features? 

We edited the Conclusion to expand its content.

Round 2

Reviewer 1 Report

the changes made to the manuscript are acceptable. congratulations to the authors